# In Vitro Tumor Cell-Binding Assay to Select High-Binding Antibody and Predict Therapy Response for Personalized ^64^Cu-Intraperitoneal Radioimmunotherapy against Peritoneal Dissemination of Pancreatic Cancer: A Feasibility Study

**DOI:** 10.3390/ijms23105807

**Published:** 2022-05-22

**Authors:** Fukiko Hihara, Hiroki Matsumoto, Mitsuyoshi Yoshimoto, Takashi Masuko, Yuichi Endo, Chika Igarashi, Tomoko Tachibana, Mitsuhiro Shinada, Ming-Rong Zhang, Gene Kurosawa, Aya Sugyo, Atsushi B. Tsuji, Tatsuya Higashi, Hiroaki Kurihara, Makoto Ueno, Yukie Yoshii

**Affiliations:** 1Department of Molecular Imaging and Theranostics, National Institutes for Quantum Science and Technology, Chiba 263-8555, Japan; hihara.fukiko@qst.go.jp (F.H.); matsumoto.hiroki2@qst.go.jp (H.M.); igarashi.chika@qst.go.jp (C.I.); tachibana.tomoko@qst.go.jp (T.T.); shinada.mitsuhiro@qst.go.jp (M.S.); zhang.ming-rong@qst.go.jp (M.-R.Z.); sugyo.aya@qst.go.jp (A.S.); tsuji.atsushi@qst.go.jp (A.B.T.); higashi.tatsuya@qst.go.jp (T.H.); 2Department of Diagnostic Radiology, Kanagawa Cancer Center, Kanagawa 241-8515, Japan; h-kurihara@kcch.jp; 3Division of Functional Imaging, National Cancer Center Hospital East, Chiba 277-8577, Japan; miyoshim@ncc.go.jp; 4School of Pharmacy, Kindai University, Osaka 577-8502, Japan; masuko@phar.kindai.ac.jp (T.M.); en-dou@phar.kindai.ac.jp (Y.E.); 5Faculty of Science, Toho University, Chiba 274-8510, Japan; 6International Center for Cell and Gene Therapy, Fujita Health University, Aichi 470-1192, Japan; gene@fujita-hu.ac.jp; 7Department of Gastroenterology, Kanagawa Cancer Center, Kanagawa 241-8515, Japan; uenom@kcch.jp

**Keywords:** ^64^Cu-intraperitoneal radioimmunotherapy, in vitro tumor cell-binding assay, pancreatic cancer, peritoneal dissemination

## Abstract

Peritoneal dissemination of pancreatic cancer has a poor prognosis. We have reported that intraperitoneal radioimmunotherapy using a ^64^Cu-labeled antibody (^64^Cu-ipRIT) is a promising adjuvant therapy option to prevent this complication. To achieve personalized ^64^Cu-ipRIT, we developed a new in vitro tumor cell-binding assay (^64^Cu-TuBA) system with a panel containing nine candidate ^64^Cu-labeled antibodies targeting seven antigens (EGFR, HER2, HER3, TfR, EpCAM, LAT1, and CD98), which are reportedly overexpressed in patients with pancreatic cancer. We investigated the feasibility of ^64^Cu-TuBA to select the highest-binding antibody for individual cancer cell lines and predict the treatment response in vivo for ^64^Cu-ipRIT. ^64^Cu-TuBA was performed using six human pancreatic cancer cell lines. For three cell lines, an in vivo treatment study was performed with ^64^Cu-ipRIT using high-, middle-, or low-binding antibodies in each peritoneal dissemination mouse model. The high-binding antibodies significantly prolonged survival in each mouse model, while low-and middle-binding antibodies were ineffective. There was a correlation between in vitro cell binding and in vivo therapeutic efficacy. Our findings suggest that ^64^Cu-TuBA can be used for patient selection to enable personalized ^64^Cu-ipRIT. Tumor cells isolated from surgically resected tumor tissues would be suitable for analysis with the ^64^Cu-TuBA system in future clinical studies.

## 1. Introduction

Pancreatic cancer has a dismal prognosis, with an overall 5-year survival rate below 10% [1,2,3,4]. Surgical resection following chemotherapy with gemcitabine is the primary treatment for patients with resectable pancreatic cancer; however, most patients subsequently experience local recurrence, hepatic metastasis, and peritoneal dissemination even after extensive surgery [5,6]. Peritoneal dissemination is observed in more than half of pancreatic cancer patients and confers a high mortality rate [5,6]. Therefore, a more effective adjuvant therapy is needed to avoid the recurrence of pancreatic cancer and improve prognosis.

To address this, we focused on intraperitoneal radioimmunotherapy using a ^64^Cu-labeled antibody (^64^Cu-ipRIT). Generally, RIT has several advantages over immunotherapy; e.g., RIT can target and kill cancer cells by irradiation from radionuclides bound to antibodies and does not require a functional immune system [7]. We developed the ^64^Cu-labeled anti-epidermal growth factor receptor (EGFR) antibody cetuximab (^64^Cu-cetuximab) to investigate the efficacy of ^64^Cu-ipRIT. Cetuximab has a high binding affinity for EGFR, which is reportedly overexpressed in >90% of pancreatic cancers [8]. ^64^Cu shows β^–^ decay (0.574 MeV, 40%), electron capture (42.6%), and β^+^ decay (0.653 MeV, 17.4%); thus, ^64^Cu can be used for internal radiotherapy, as well as for positron emission tomography (PET) imaging. For therapeutic use, β^–^ particles and Auger electrons emitted from ^64^Cu damage tumor cells, with the high-linear energy transfer Auger electrons causing heavy damage to cancer cell DNA [9,10]. The ipRIT with ^64^Cu-cetuximab was shown to be effective in inhibiting local recurrence and regrowth of distant metastasis, including peritoneal dissemination and liver metastasis. It also significantly prolonged survival with little toxicity, as observed using an orthotopic xenograft mouse model after surgery to resect primary pancreatic tumors [11,12]. However, patients with weak or moderate expression of EGFR should be administered more effective ^64^Cu-ipRIT using another high-binding ^64^Cu-labeled antibody. To accomplish this personalized approach for ^64^Cu-ipRIT, the selection of a high-binding antibody for individual cancers prior to treatment is important.

Western blotting has been widely used as a method to investigate the relative protein expression levels of a certain target antigen and compare these among individual tumors [13,14]. However, it is typically considered that Western blot can only provide semi-quantitative analysis to compare expression levels of multiple target antigens. This is due to the unavoidable variations among separate blots and differences in specificity among different antibodies to target different proteins [15,16]. In previous studies on radioimmunotherapy, cell-binding assays have been used as a useful technique to investigate the binding affinity of radiolabeled antibodies and to estimate tumor uptake in vivo [17].

Recently, in vitro assays with tumor cells or primary tumor cell cultures, obtained from individual resected tumor tissues, have been actively studied, and their usefulness as a tool for selecting optimal drugs or antibodies in personalized chemotherapy or molecular-targeted therapy has also been evaluated [18]. Therefore, we hypothesized that an in vitro cell-binding assay would be a possible tool for selecting high-binding antibodies from multiple candidate antibodies to achieve personalized ^64^Cu-ipRIT. However, the correlation between in vitro antibody binding and in vivo therapeutic efficacy for ^64^Cu-ipRIT remains unclear.

Here, we investigated the feasibility of using an in vitro tumor cell-binding assay to select the optimal antibody for an individual cancer and to predict treatment response in vivo, using human pancreatic cancer cell lines. As a proof-of-concept, we developed a new in vitro tumor cell-binding assay system, the ^64^Cu-TuBA system, using a panel containing nine candidate ^64^Cu-labeled antibodies: ^64^Cu-anti-EGFR antibodies (cetuximab and panitumumab), anti-HER2 antibodies (trastuzumab and pertuzumab), anti-HER3, anti-TfR, anti-EpCAM, anti-LAT1, and anti-CD98 antibodies (Table 1). To establish the antibody panel for ^64^Cu-TuBA, we selected seven antigens (EGFR, HER2, HER3, TfR, EpCAM, LAT1, and CD98), which are reportedly overexpressed in pancreatic cancer patients [8,19,20,21,22,23,24] (Table 1). This proof-of-concept study used the same ^64^Cu-labeled antibodies to perform the in vitro tumor cell-binding assay as those used in ^64^Cu-ipRIT because this would result in a better prediction of in vivo efficacy by in vitro assay.

## 2. Results

### 2.1. In Vitro Tumor Cell-Binding Assay with ^64^Cu-Labeled Antibodies

The schematic of this study is shown in Figure 1. We generated a panel containing seven candidate ^64^Cu-labeled antibodies, including ^64^Cu-anti-EGFR antibodies (cetuximab and panitumumab), anti-HER2 antibodies (trastuzumab and pertuzumab), anti-HER3 antibodies, anti-TfR antibodies, anti-EpCAM antibodies, anti-LAT1 antibodies, and anti-CD98 antibodies. We then conducted an in vitro cell-binding assay with this ^64^Cu-labeled antibody panel with six human pancreatic cancer cell lines (AsPC-1, BxPC-3, Capan-1, MIA PaCa-2, PANC-1, and PSN-1) (Figure 2). The ^64^Cu-anti-EGFR antibody cetuximab showed high cell binding in AsPC-1, BxPC-3, MIA PaCa-2, PANC-1, and PSN-1 (44.7%, 32.4%, 32.9%, 51.2%, and 31.2%), whereas Capan-1 showed low cell binding (9.2%). There were significant differences between these cell lines (AsPC-1, BxPC-3, MIA PaCa-2, PANC-1, PSN-1, vs. Capan-1, respectively) (*p* < 0.05). ^64^Cu-cetuximab showed higher cell binding than ^64^Cu-panitumumab in all cell lines (*p* < 0.05). Capan-1 showed higher cell binding for ^64^Cu-anti-TfR antibody (23.2%) than ^64^Cu-cetuximab (9.2%) (*p* < 0.05). The other ^64^Cu-labeled antibodies, ^64^Cu-anti-HER2 antibodies (trastuzumab and pertuzumab), anti-HER3, anti-TfR, anti-EpCAM, anti-LAT1, and anti-CD98 antibodies showed low cell binding in all cell lines. In addition, we observed a correlation between EGFR expression as measured by Western blotting and cell binding (%) (*R* = 0.837, *p* = 0.0377), but the coefficient of variation (CV) of Western blotting was significantly greater than that of cell binding (%) (Figure 3) (*p* < 0.05). Thus, the in vitro cell-binding assay showed smaller variations than Western blotting.

### 2.2. In Vivo ^64^Cu-ipRIT Study Using Peritoneal Dissemination Models

For the in vivo study, peritoneal dissemination mouse models with three pancreatic cancer cell lines, AsPC-1, Capan-1, and PSN-1, were used. The efficacy of ^64^Cu-ipRIT was examined in vivo using representative ^64^Cu-labeled antibodies, including ^64^Cu-anti-EGFR antibody (cetuximab), ^64^Cu-anti-TfR antibody, and ^64^Cu-anti-CD98 antibody, and was determined to be high, medium, or low binding for each cell line. The survival curves are shown in Figure 4. In peritoneal dissemination mouse models using AsPC-1, the survival after ^64^Cu-ipRIT with ^64^Cu-anti-EGFR antibody (cetuximab), showing high binding for AsPC-1 cells, was greater than that observed for the saline control (*p* = 0.0025). On the other hand, there were no significant differences in survival after ^64^Cu-ipRIT with ^64^Cu-anti-TfR antibody and ^64^Cu-anti-CD98 antibody, showing moderate and low binding for AsPC-1 cells compared to the saline control. For the peritoneal dissemination mouse models of PSN-1, survival after ^64^Cu-ipRIT with ^64^Cu-anti-EGFR antibody (cetuximab) and ^64^Cu-anti-TfR antibody, showing high and middle binding for PSN-1 cells, was greater than that observed for the saline control (*p* = 0.0004 and 0.0081, respectively). However, no significant difference was found for the ^64^Cu-anti-CD98 antibody, showing low binding for PSN-1 cells compared to the saline control. The peritoneal dissemination mouse models of Capan-1 with ^64^Cu-anti-TfR antibody showed high binding for Capan-1 cells, which was higher than that of the saline control (*p* = 0.0142). Comparatively, no significant difference was detected in ^64^Cu-cetuximab and ^64^Cu-anti-CD98 antibodies, showing middle-low binding for Capan-1 cells compared with the saline control. The mean survival time (MST) and %MST are summarized in Table 2. The MST values were as follows: ^64^Cu-cetuximab > ^64^Cu-anti-TfR antibody > ^64^Cu-anti-CD98 antibody > saline control in AsPC-1 and PSN-1, while the values were ^64^Cu-anti-TfR antibody > ^64^Cu-cetuximab > ^64^Cu-anti-CD98 antibody > saline control in Capan-1. In the in vivo treatment study, all mice in each group reached a humane endpoint due to noticeable extension of the abdomen due to tumor growth in the peritoneum. After treatment, there was no weight loss of more than 20% compared with the initial body weight due to drug administration in any treatment group for all cell line models (Appendix A). Figure 5 shows the correlation between cell binding (%) from the in vitro study and relative survival time from the in vivo study in each cell line. Strong correlations were observed in all examined cell lines (*R* = 0.9999, *p* = 0.0072 in AsPC-1; *R* = 0.9971, *p* = 0.0479 in PSN-1; *R* = 0.9972, *p* = 0.0478 in Capan-1).

## 3. Discussion

We demonstrated that ^64^Cu-TuBA systems selected high-binding antibodies for individual cancer cell lines and predicted treatment response in each peritoneal dissemination mouse model. These findings indicated the feasibility of ^64^Cu-TuBA and suggested that this method would be useful to enable personalized ^64^Cu-ipRIT. Pancreatic cancer has one of the poorest prognoses among all types of cancer [1,2,3,4,5]. We previously demonstrated that ipRIT with ^64^Cu-cetuximab was effective as adjuvant therapy after surgery for pancreatic cancer in vivo [11]. Assuming the possible clinical workflow for the personalized ^64^Cu-ipRIT for use as a postoperative adjuvant therapy, an in vitro tumor cell-binding assay can be conducted with tumor cells obtained from isolated pancreatic cancer samples immediately after surgery. Then, based on the in vitro tumor cell-binding assay with individual patient tumor cells, a high-binding ^64^Cu-labeled antibody can be selected and used for the ^64^Cu-ipRIT adjuvant therapy. Since ^64^Cu-TuBA was able to select high-binding ^64^Cu-labeled antibodies in all the examined pancreatic cancer cell lines, this method should provide optimal antibodies to most pancreatic cancer patients.

We used a panel containing nine candidate ^64^Cu-labeled antibodies targeting seven antigens as a proof-of-concept study of ^64^Cu-TuBA. Of the examined antibodies, those for EGFR and TfR showed relatively high binding to the pancreatic cancer cells used in this study, compared with those for HER2, HER3, EpCAM, LAT1, and CD98. Our results suggest that this assay can be easily applied to a variety of antibodies. For the future clinical use of this assay, it would be beneficial to add the other candidate antibodies used in the panel. Antibody arrays, which have been variously developed for biomarker detection in pancreatic cancer [36,37], might be applied in the ^64^Cu-TuBA format in the future. Thus far, no pancreatic cancer-specific therapeutic antibodies have been approved for the treatment of pancreatic cancer despite the effort of previous studies; that is, it is difficult to cover most pancreatic cancer patients by targeting one specific target antibody [38]. Therefore, personalized ^64^Cu-ipRIT with ^64^Cu-TuBA could be a beneficial strategy for pancreatic cancer treatment. We used ^64^Cu-labeled antibodies, rather than antibodies with other types of labels, for the in vitro tumor cell-binding assays as a proof-of-concept study, since these are the same compounds used in ^64^Cu-ipRIT. This study successfully demonstrated the correlation between binding ability in in vitro cell-binding assays with ^64^Cu-labeled antibodies and in vivo therapeutic efficacy of ^64^Cu-ipRIT. Thus, this is an important basic finding to advance further development of personalized ^64^Cu-ipRIT. To make this assay easier to use in various locations, replacing ^64^Cu-labeled antibodies with fluorescent-labeled antibodies might be worth exploring in future studies. However, it is necessary to note the differences between ^64^Cu-labeling and fluorescent-labeling, such as binding affinity and labeling efficiency.

From the in vivo treatment study, we found that there was a strong correlation between cell binding (%) from the in vitro study and the relative survival time in all examined cell line models. These findings suggest the feasibility of an in vitro tumor cell-binding assay to observe not only cell binding, but also predict the therapeutic efficacy in vivo. We observed that ^64^Cu-ipRIT with ^64^Cu-cetuximab was effective in AsPC-1 and PSN-1, while ^64^Cu-anti-TfR antibody was effective for Capan-1 in each peritoneal dissemination mouse model. In these cases, cell binding (%) showed 44.7 ± 3.0% and 31.2 ± 0.3% for ^64^Cu-cetuximab in AsPC-1 and PSN-1 and 23.2 ± 1.1% for ^64^Cu-anti-TfR antibody in Capan-1. Thus, it could be effective in vivo, at least when cell binding (%) is more than 20%, as in this experiment. In this in vivo experiment, we observed significant increases in survival time with respective treatment with high-binding antibodies for each cell line model. However, all models subsequently recurred. In our previous study, we showed that vorinostat, a histone deacetylase inhibitor, is an effective radiosensitizer for use in the treatment of peritoneal dissemination of gastric cancer by ipRIT with ^64^Cu-cetuximab [39]. This suggests that the combined use of vorinostat has the potential to facilitate the efficacy of ^64^Cu-ipRIT for the treatment of peritoneal dissemination of pancreatic cancer.

This study has several limitations. First, the present study used one fixed administration dose for the in vivo treatment of ^64^Cu-ipRIT (22.2 MBq/mouse), which was determined by previous studies with ^64^Cu-cetuximab, for comparison. This was optimized as the maximum tolerated dose of ^64^Cu-cetuximab, which was the best antibody for in vivo treatment in the present study. The optimal doses for each ^64^Cu-labeled antibody will be evaluated in future clinical trials. Second, in the present study, we used peritoneal dissemination mouse models for the in vivo studies. In clinical practice, most pancreatic cancer patients show other types of recurrence, such as local recurrence and hepatic metastasis, as well as peritoneal dissemination after surgery [1,5,6]. Although our previous study demonstrated that ^64^Cu-ipRIT with ^64^Cu-cetuximab reduces local recurrence and hepatic metastasis [11], it is necessary to investigate the efficacy of the other ^64^Cu-labeled antibodies against other types of recurrence in future preclinical and clinical studies.

## 4. Materials and Methods

### 4.1. Preparation of ^64^Cu-Labeled Antibody and ^64^Cu-Labeled Antibody Panel

Antibodies used in this study and the sources are listed in Table 1. ^64^Cu was produced on a cyclotron at the National Institutes for Quantum and Radiological Science and Technology (QST, Chiba, Japan) and purified using previously published methods [40]. According to our previous study [12], the bifunctional chelator p-SCN-Bn-PCTA (Macrocyclics) was used for antibody conjugation. The ^64^Cu-PCTA-antibodies were prepared using methods reported in a previous study [12], with a specific activity of 1.7 GBq/mg. A ^64^Cu-labeled antibody panel was prepared before the in vitro cell-binding assay, as shown in Figure 1, with 2 mL centrifuge tubes.

### 4.2. Cell Culture

Human pancreatic cancer cell lines (AsPC-1, BxPC-3, Capan-1, MIA PaCa-2, PANC-1, and PSN-1) were obtained from the American Type Culture Collection (ATCC, Manassas, VA, USA). The cells were cultured in a humidified atmosphere of 5% CO_2_ at 37 °C. RPMI 1640 (Wako) supplemented with 10% fetal bovine serum (FBS) was used for AsPC-1, BxPC-3, and PSN-1. On the other hand, DMEM (Wako, Osaka, Japan) supplemented with 10% FBS was used for MIA PaCa-2 and PANC-1, while IMDM (Gibco, Waltham, MA, USA) supplemented with 20% FBS was used for Capan-1. Exponentially growing cells were detached from the culture plates with trypsin and used in this study. The number of viable cells was determined using the trypan blue dye exclusion method.

### 4.3. In Vitro Tumor Cell-Binding Assay with ^64^Cu-Labeled Antibodies

An in vitro cell-binding assay was performed using the ^64^Cu-labeled antibody panel. For the cell-binding assay, 3 × 10^5^ cultured cells from each cell line were diluted in 1 mL of ice-cold phosphate-buffered saline (PBS) with 1% bovine serum albumin (BSA) (Sigma-Aldrich) and were added to 2 mL centrifuge tubes and incubated with each ^64^Cu-labeled antibody (20 kBq) on ice for 1 h. Then cells were washed with ice-cold PBS on ice. After washing, the radioactivity bound to the cells was measured using a γ-counter (1480 Automatic Gamma Counter Wizard 3; PerkinElmer). The percentage of cell binding was calculated as (radioactivity of the collected cells/radioactivity administered to the cells × 100) (%).

### 4.4. Western Blot Analysis

Protein expression levels of EGFR were examined by Western blot analysis and compared with values of the cell binding (%) obtained from the cell-binding assay in each cell line. The cultured cells (3 × 10^5^ cells, *n* = 3 for each cell line) were lysed with lysis buffer containing protease inhibitor cocktail (Sigma-Aldrich, Burlington, MA, USA) according to the manufacturer’s protocol, and protein concentrations were determined using a BCA protein assay kit (ThermoFisher Scientific, Waltham, MA, USA). The SDS-polyacrylamide gel electrophoresis was performed with 15 μg of protein in each sample using 5–20% gel (ATOO) and transferred to a PVDF membrane (BioRad, Hercules, CA, USA). After blocking in 0.05% Triton-TBS containing 1% BSA (NACALAI TESQUE, INC., Kyoto, Japan) at room temperature for 30 min, the membrane was incubated at 4 °C overnight with each primary antibody. For the primary antibodies, rabbit anti-EGFR antibodies (4267, Cell Signaling Technology) and mouse anti-GAPDH antibodies (MCA4739, AbD Serotec, Oxford, UK) were used as loading controls. Then, the excess antibody was washed with 0.05% TBST, and the membrane was incubated with the secondary antibody (HRP-linked anti-rabbit or mouse IgG antibody (7074S, Cell Signaling Technology, Danvers, MA, USA)) at room temperature for 2 h. After washing, the membrane was incubated with SuperSignal West Pico Chemiluminescent Substrate (ThermoFisher Scientific, Waltham, MA, USA) at room temperature for 5 min, and signals were detected using an X-ray film. After exposure, the membrane was incubated in stripping buffer at 37 °C for 30 min to strip off the former antibody. The intensity was calculated by densitometry using the ImageJ software (National Institutes of Health).

### 4.5. Animal Experiments

Six-week-old female NOD.CB17-Prkdc SCID/J mice (SCID mice, 15–20 g bodyweight) were obtained from Charles River Laboratories (Yokohama, Japan) and were used in this study. Before the experiments, the mice were acclimated for at least 1 week. All animal experimental procedures were approved by the Animal Ethics Committee of the National Institutes for Quantum Science and Technology and conducted in accordance with the institutional guidelines. To generate peritoneal dissemination mouse models, AsPC-1, Capan-1, and PSN-1 cell lines were used. Cells (5 × 10^6^) suspended in 500 µL phosphate buffered saline (PBS) were injected intraperitoneally 1 week before treatment which resulted in the small nodules of peritoneal dissemination 1 week after cell inoculation.

### 4.6. In Vivo Treatment Study of ^64^Cu-ipRIT Using the Peritoneal Dissemination Models

The efficacy of ^64^Cu-ipRIT was examined in vivo with representative ^64^Cu-labeled antibodies, including ^64^Cu-anti-EGFR antibody (cetuximab), ^64^Cu-anti-TfR antibody, and ^64^Cu-anti-CD98 antibody using peritoneal dissemination mouse models of AsPC-1, Capan-1, and PSN-1 cell lines. Mice with peritoneal dissemination were randomized into four groups for each cell line (*n* = 7/group). Mice were injected intraperitoneally with 22.2 MBq ^64^Cu-anti-EGFR antibody (cetuximab), ^64^Cu-anti-TfR antibody, or ^64^Cu-anti-CD98 antibody (day 0; 7days after cell inoculation) (^64^Cu-anti-EGFR antibody group, ^64^Cu-anti-TfR antibody group, ^64^Cu-anti-CD98 antibody group, respectively). For comparison, mice were examined after administration of saline (day 0; 7 days after cell inoculation) (saline control group). The dose of ^64^Cu-ipRIT was determined based on a previous report [12]. The mice were weighed and observed thereafter. The humane endpoint was defined as a noticeable extension of the abdomen, development of ascites, or bodyweight loss (>20%). Mean survival time (MST) was determined, and the percentage of increase in MST (treatment) was calculated as (MST of treatment group/MST of the saline control group × 100) (%). To compare in vivo treatment efficacy with in vitro cell binding, the relative survival time was calculated as (survival time for each mouse/average survival time for each saline control).

### 4.7. Statistical Analysis

Data were expressed as means with corresponding standard deviations. Multiple comparisons were conducted using one-way analysis of variance (ANOVA) with post hoc comparisons using the Tukey–Kramer test. Differences in survival were evaluated using log-rank tests. Statistical significance was set at *p* < 0.05.

## 5. Conclusions

This study demonstrated the feasibility of an in vitro tumor cell-binding assay, ^64^Cu-TuBA, to select antibodies with high binding affinity for individual cancer and to predict treatment response to ^64^Cu-ipRIT. This method would enable individual patients with pancreatic cancer to receive the optimal treatment, leading to better patient care and lower costs.

## Figures and Tables

**Figure 1 ijms-23-05807-f001:**
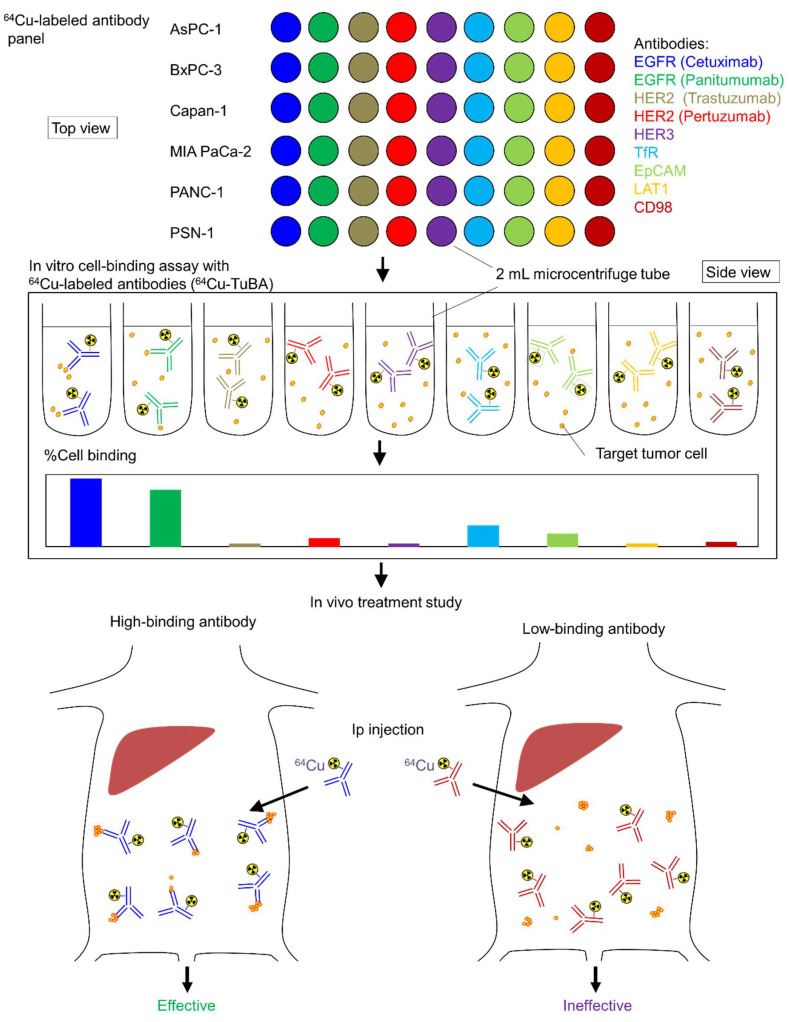
Scheme of in vitro tumor cell-binding assay (^64^Cu-TuBA) and personalized ^64^Cu-intraperitoneal radioimmunotherapy (^64^Cu-ipRIT).

**Figure 2 ijms-23-05807-f002:**
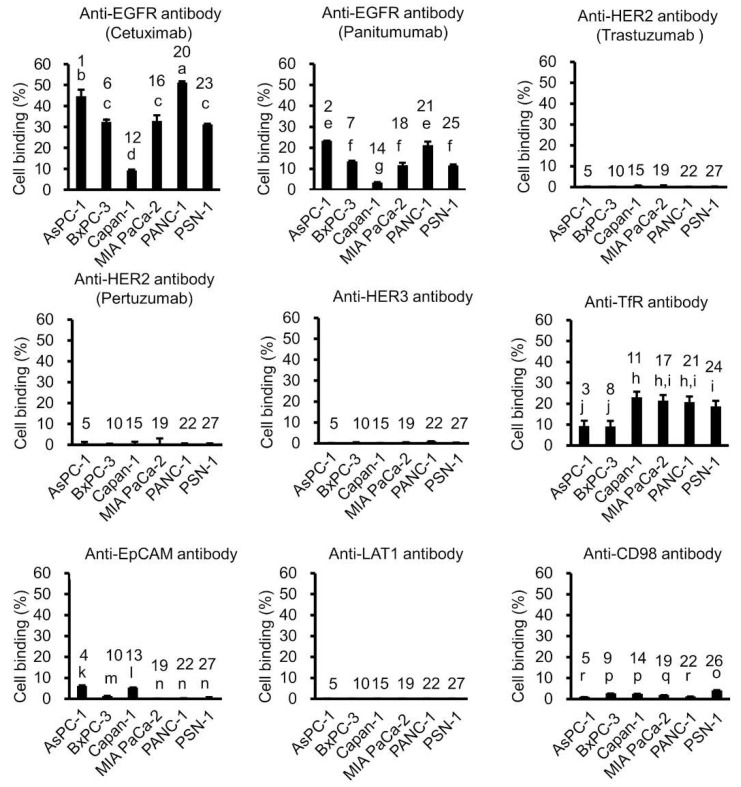
In vitro tumor cell-binding assay with ^64^Cu-labeled antibodies. Cell binding (%) for nine candidate ^64^Cu-labeled antibodies, including ^64^Cu-anti-EGFR antibodies (cetuximab and panitumumab), anti-HER2 antibodies (trastuzumab and pertuzumab), anti-HER3, anti-TfR, anti-EpCAM, anti-LAT1, and anti-CD98 antibodies in six human pancreatic cancer cell lines (AsPC-1, BxPC-3, Capan-1, MIA PaCa-2, PANC-1, and PSN-1). There were significant differences between different characters in a–c; e–g; h–j; k–n; o–r, among antibodies; 1–5; 6–10; 11–15; 16–19; 20–22; 23–27, and among the cell lines, respectively (*p* < 0.05).

**Figure 3 ijms-23-05807-f003:**
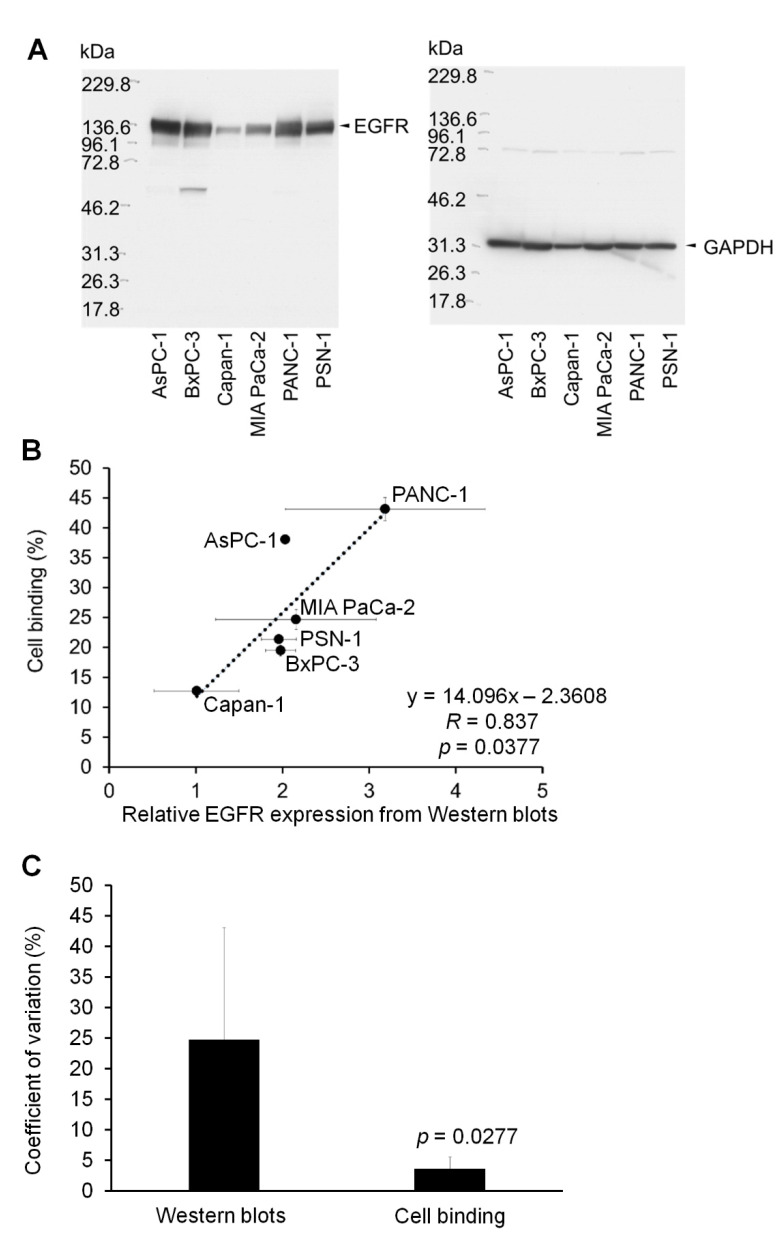
Relationships between Western blots for EGFR expression and in vitro cell-binding assay with ^64^Cu-anti-EGFR antibodies (cetuximab). (**A**) Representative images of Western blots for EGFR and GAPDH expression. (**B**) Correlation between relative EGFR expression (EGFR/GAPDH) from Western blots and cell binding (%) from an in vitro cell-binding assay with ^64^Cu-anti-EGFR antibodies (cetuximab). (**C**) Coefficient of variation (%) for relative EGFR expression from Western blots (Western blots) and cell binding (%) from an in vitro cell-binding assay (cell binding).

**Figure 4 ijms-23-05807-f004:**
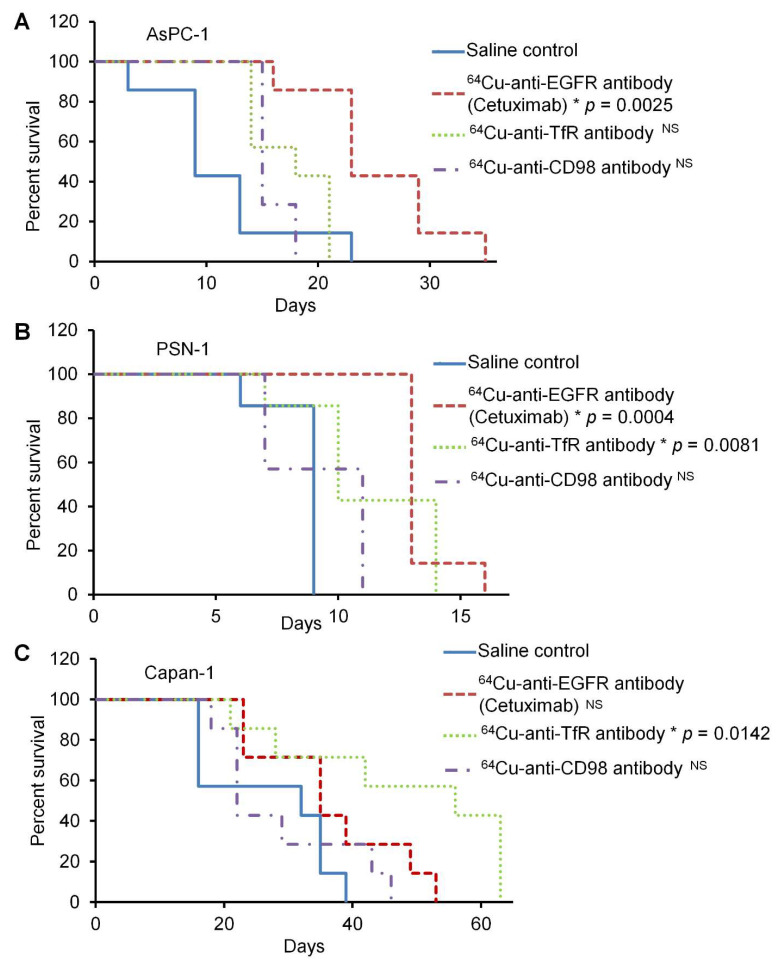
In vivo ^64^Cu-i*p*RIT study using the peritoneal dissemination models of AsPC-1, PSN-1, and Capan-1. Survival curves of the saline control (blue line), ^64^Cu-anti-EGFR antibody (cetuximab) (red line), ^64^Cu-anti-TfR antibody (green line), and ^64^Cu-anti-CD98 antibody (purple line) for AsPC-1 (**A**), PSN-1 (**B**), and Capan-1 (**C**), respectively (*n* = 7/group). Asterisks indicate significant differences (*p* < 0.05). NS = not significant.

**Figure 5 ijms-23-05807-f005:**
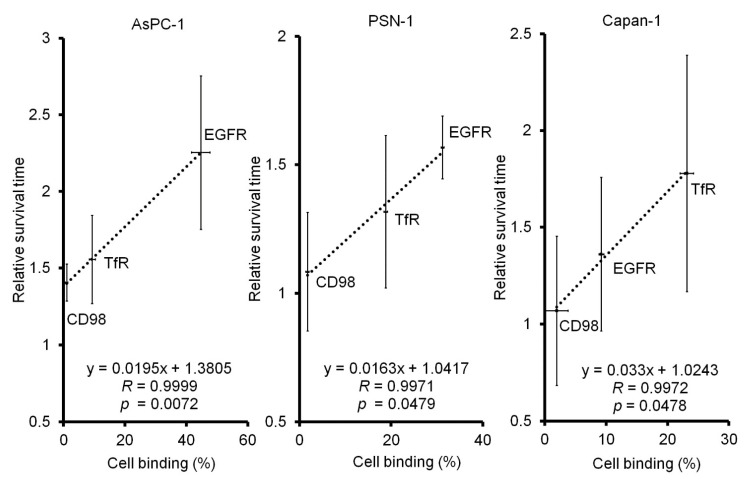
Relationships between in vitro cell-binding assay and in vivo treatment study. Correlation between cell binding (%) from the in vitro study and relative survival time from the in vivo study in AsPC-1 (**left**), PSN-1 (**middle**), and Capan-1 (**right**).

**Table 1 ijms-23-05807-t001:** Target antigens and antibodies used in the ^64^Cu-TuBA assay system.

Antigens	Abbreviations	Antibodies	Source
Epidermal growth factor receptor	EGFR	cetuximab	Merck Serono
		panitumumab	Takeda
Human epidermal growth factor receptor 2	HER2	trastuzumab	Chugai Pharmaceutical
		pertuzumab	Chugai Pharmaceutical
Human epidermal growth factor receptor 3	HER3	Ab3-1	[25,26,27]
Transferrin receptor	TfR	066-188	[28,29,30]
Epithelial cell adhesion molecule	EpCAM	1D12	[31]
L-type amino acid transporter 1	LAT1	Ab1	[32,33,34]
4F2 heavy chain	CD98	HBJ127	[35]

**Table 2 ijms-23-05807-t002:** Mean survival time (MST) from in vivo treatment study in the peritoneal dissemination mouse models of AsPC-1, PSN-1, and Capan-1 cell lines.

Groups	AsPC-1			
	Mean Survival Time			%MST
Saline control	11	±	6	100
^64^Cu-cetuximab	25	±	6	225
^64^Cu-anti-TfR antibody	18	±	3	156
^64^Cu-anti-CD98 antibody	16	±	1	141
**Groups**	**PSN-1**			
	**Mean survival time**			**%MST**
Saline control	9	±	1	100
^64^Cu-cetuximab	13	±	1	157
^64^Cu-anti-TfR antibody	11	±	3	132
^64^Cu-anti-CD98 antibody	9	±	2	108
**Groups**	**CAPAN-1**			
	**Mean survival time**			**%MST**
Saline control	27	±	10	100
^64^Cu-cetuximab	37	±	11	136
^64^Cu-anti-TfR antibody	48	±	16	178
^64^Cu-anti-CD98 antibody	29	±	10	107

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
