# Peer review of "In Vitro Tumor Cell-Binding Assay to Select High-Binding Antibody and Predict Therapy Response for Personalized 64Cu-Intraperitoneal Radioimmunotherapy against Peritoneal Dissemination of Pancreatic Cancer: A Feasibility Study"

_ijms, 2022, doi:10.3390/ijms23105807_

Round 1

Reviewer 1 Report

Hihara et al. describe using a cell binding assay to predict in vivo efficacy of antibody-Cu64 conjugates. However, there are numerous questions remaining, particularly that Figures 1 and 2 are missing from the draft.

Concerns/Questions:

  • Introduction - For the last sentence of the Introduction, it says that the study used “Cu64-labeled antibodies . . . because these have the same structure as those used for 64Cu-ipRIT. Are the Cu64-labeled antibodies being used not the 64Cu-ipRIT? (As opposed to “the same” as the 64Cu-ipRIT.)
  • Figures 1 & 2 are missing. – This makes it impossible to evaluate the claims of the paper.
  • 3, line 122 – The way it is written makes it sound like CV is larger for the cell binding experiments instead of the opposite.
  • Figure 3B – What do the dots on the graph represent? I assume they are the different cell lines (?). If so, please delineate which dot corresponds to which cell line.
  • Figure 5 – This figure uses the term HER1 instead of EGFR. The authors should be consistent with nomenclature.
  • Discussion – Lines 245-246 – This statement seems like too broad of a conclusion. It would be better to say that cell binding > 20% correlated to in vivo efficacy for this particular experiment rather than drawing the broad that conclusion that this will always be the case.
  • Discussion – Lines 256-257 – This sentence doesn’t make sense.
  • Materials & Methods – Line 290 – How were the cells washed? Were they kept on ice during the washing steps?
  • Materials & Methods – Under the “Animal experiments” section, it says that the cell lines were injected 1 week prior to treatment. However, under the “In vivo treatment” section, it says that mice were injected with the antibodies on day 0. Please clarify the timeline. Were the mice verified to have tumors prior to treatment with the antibodies?
  • Materials & Methods – Line 335 – States that bodyweight loss was one of the reasons for mouse sacrifice, but in the text, the authors stated that no mice were sacrificed due to weight loss.
  • Figure S1 – These bands look different than the ones shown in Figure 3 (for both the EGFR and GAPDH bands). Have these images been broadened or the images in Figure 3A been squished? In particular, Figure S1A makes it look like there are 2 different EGFR bands while there only appears to be 1 EGFR band in Figure 3.

Author Response

Reviewer’s general comments: Hihara et al. describe using a cell binding assay to predict in vivo efficacy of antibody-Cu64 conjugates. However, there are numerous questions remaining, particularly that Figures 1 and 2 are missing from the draft.

Response:

Thank you very much for your review. Our submitted word file contained Figures 1 and 2, but unfortunately Figures 1 and 2 were missing in the PDF file provided to the reviewers. We sincerely apologize for the missing Figures. We have revised our manuscript according to your insightful suggestions, as detailed below.

Introduction - For the last sentence of the Introduction, it says that the study used “Cu64-labeled antibodies . . . because these have the same structure as those used for 64Cu-ipRIT. Are the Cu64-labeled antibodies being used not the 64Cu-ipRIT? (As opposed to “the same” as the 64Cu-ipRIT.)

Response:

As the reviewer pointed out, this sentence was confusing. To clarify this point, we revised the sentence as follows.

P2L97: This proof-of-concept study used the same 64Cu-labeled antibodies to perform the in vitro tumor cell-binding assay as those used in 64Cu-ipRIT because this would result in a better prediction of in vivo efficacy by in vitro assay.

Figures 1 & 2 are missing. – This makes it impossible to evaluate the claims of the paper.

Response:

We sincerely apologize that Figures 1 & 2 were missing We ensured that Figures 1 & 2 are provided in the resubmitted PDF file.

3, line 122 – The way it is written makes it sound like CV is larger for the cell binding experiments instead of the opposite.

Response:

Thank you very much for your insightful comment. To improve clarification, we revised the content as follows:

P3L121: In addition, we observed a correlation between EGFR expression as measured by western blotting and cell binding (%) (R = 0.837, P = 0.0377), but the coefficient of variation (CV) of western blotting was significantly greater than that of cell binding (%) (Figure 3,) (P < 0.05).

Figure 3B – What do the dots on the graph represent? I assume they are the different cell lines (?). If so, please delineate which dot corresponds to which cell line.

Response:

Thank you very much for your helpful comment. We added the cell line names to Figure 3B.

Figure 5 – This figure uses the term HER1 instead of EGFR. The authors should be consistent with nomenclature.

Response:

We sincerely apologize for this inconsistency. We revised HER1 to EGFR in Figure 5.

Discussion – Lines 245-246 – This statement seems like too broad of a conclusion. It would be better to say that cell binding > 20% correlated to in vivo efficacy for this particular experiment rather than drawing the broad that conclusion that this will always be the case.

Response:

As the reviewer kindly suggested, this statement was too broad. Accordingly, we revised this sentence to describe the finding of this experiment as follows:

P10L251: Thus, it could be effective in vivo, at least when cell binding (%) is more than 20%, as in this experiment.

Discussion – Lines 256-257 – This sentence doesn’t make sense.

Response:

Thank you for your kind comment. To clarify the meaning, we revised the sentence as follows:

P11L263; The optimal doses for each 64Cu-labeled antibody will be evaluated in future clinical trials.

Materials & Methods – Line 290 – How were the cells washed? Were they kept on ice during the washing steps?

Response:

According to your kind suggestion, we revised here as follows.

P11L292; An in vitro cell-binding assay was performed using the 64Cu-labeled antibody panel. For the cell-binding assay, 3 ´ 105 cultured cells from each cell line were diluted in 1 mL of ice-cold phosphate-buffered saline (PBS) with 1% bovine serum albumin (BSA) (Sigma) and were added to 2 mL centrifuge tubes and incubated with each 64Cu-labeled antibody (20 kBq) on ice for 1 h. Then cells were washed with ice-cold PBS on ice. After washing, the radioactivity bound to the cells was measured using a γ-counter (1480 Automatic Gamma Counter Wizard 3; PerkinElmer).

Materials & Methods – Under the “Animal experiments” section, it says that the cell lines were injected 1 week prior to treatment. However, under the “In vivo treatment” section, it says that mice were injected with the antibodies on day 0. Please clarify the timeline. Were the mice verified to have tumors prior to treatment with the antibodies?

Response:

According to the reviewer’s kind suggestion, we revised the following sentences to clarify the timeline and the state of the models.

P12L327; Cells (5 × 106 ) suspended in 500 µL phosphate buffered saline (PBS) were injected intraperitoneally 1 week before treatment which resulted in the small nodules of peritoneal dissemination 1 week after cell inoculation.

P12L336; Mice were injected intraperitoneally with 22.2 MBq 64Cu-anti-EGFR antibody (cetuximab), 64Cu-anti-TfR antibody, or 64Cu-anti-CD98 antibody (day 0; 7days after cell inoculation) (64Cu-anti-EGFR antibody group, 64Cu-anti-TfR antibody group, 64Cu-anti-CD98 antibody group, respectively). For comparison, mice were examined after administration of saline (day 0; 7days after cell inoculation) (saline control group).

Materials & Methods – Line 335 – States that bodyweight loss was one of the reasons for mouse sacrifice, but in the text, the authors stated that no mice were sacrificed due to weight loss.

Response:

As the reviewer kindly suggested, the sentence in the Method section was confusing, although we wanted to state the definition of the human endpoint in in vivo experiment. To clarify this, we revised the sentence as follows.

P12L342; The humane endpoint was defined as a noticeable extension of the abdomen, development of ascites, or body weight loss (>20%).

Figure S1 – These bands look different than the ones shown in Figure 3 (for both the EGFR and GAPDH bands). Have these images been broadened or the images in Figure 3A been squished? In particular, Figure S1A makes it look like there are 2 different EGFR bands while there only appears to be 1 EGFR band in Figure 3.

Response:

In the original manuscript, the bands in Figure 3A and Figure S1 were the same, but those in Figure 3A were squished, as the reviewer suggested. In the original manuscript, we represented Figure S1 as the original figure, according to the Journal’s request. However, this was redundant. So, we incorporated Figure S1 into Figure 3A in the revised manuscript.

Reviewer 2 Report

This is a well written paper which describes an interesting approach to assessment of RIT.

It seems that Figures 1 and 2 are missing from the file I received. These present the in vitro data.

MINOR

The main limitation of the study is the use of a single administered activity. However, activity can be optimized once the best antibody has been selected.

TYPOS ETC

Page 2, line 87: in vivo should be in italics

Author Response

Reviewer’s general comments: This is a well written paper which describes an interesting approach to assessment of RIT. It seems that Figures 1 and 2 are missing from the file I received. These present the in vitro data.

Response:

Thank you very much for your review. Our submitted word file contained Figures 1 and 2, but unfortunately Figures 1 and 2 were missing in the PDF file provided to the reviewers. We sincerely apologize for the missing Figures. We confirmed that Figures 1 & 2 are found in the resubmitted PDF file. We have revised our manuscript according to your insightful suggestions, as detailed below.

MINOR The main limitation of the study is the use of a single administered activity. However, activity can be optimized once the best antibody has been selected.

Response:

According to the reviewer’s kind comment, we revised the description of the limitations of this study, as follows.

P11L259; This study has several limitations. First, the present study used one fixed administration dose for the in vivo treatment of 64Cu-ipRIT (22.2 MBq/mouse), which was determined by previous studies with 64Cu-cetuximab, for comparison. This was optimized as the maximum tolerated dose of 64Cu-cetuximab, which was the best antibody for in vivo treatment in the present study. The optimal doses for each 64Cu-labeled antibody will be evaluated in future clinical trials.

TYPOS ETC

Page 2, line 87: in vivo should be in italics

Response: As the reviewer pointed out, we revised here, as follows.

P2L85; However, the correlation between in vitro antibody binding and in vivo therapeutic efficacy for 64Cu-ipRIT remains unclear.
